# Broadband Frequency Selective Rasorber Based on Spoof Surface Plasmon Polaritons

**DOI:** 10.3390/mi13111969

**Published:** 2022-11-13

**Authors:** Jin Bai, Qingzhen Yang, Yichao Liang, Xiang Gao

**Affiliations:** 1School of Power and Energy, Northwestern Polytechnical University, Xi’an 710129, China; 2School of Aerospace Engineering, Xiamen University, Xiamen 361005, China

**Keywords:** frequency selective rasorber (FSR), spoof surface plasmon polaritons (SSPP), frequency selective surface (FSS), broadband

## Abstract

A broadband frequency selective rasorber (FSR) based on spoofsurface plasmon polaritons (SSPP) is proposed. The FSR is composed of a multi-layer structure comprising frequency selective surface (FSS)-polyresin (PR)-indium tin oxide (ITO)-PR-FSS and placed vertically on a metal base plate. A periodic square cavity structure is formed. The transmission characteristics of the FSR are studied by full-wave simulation and equivalent circuit method. The simulation results demonstrate that under normal incidence, the absorption rate of the structure remains 95% in the 5–30 GHz band, and the absorption rate is also 80% in the 3.5–5 GHz band. As the incident angle of the electromagnetic wave increases to 40°, the absorption rate in the 15–20 GHz band decreases to 70% in the transverse electric (TE) mode, and the absorption rate in the transverse magnetic (TM) mode is almost the same as that of vertical incidence. The transmission response of the structure is measured in an anechoic chamber. The measurement results agree well with the simulation results, proving the reliability of the design and fabrication. The structure is less sensitive to the incident angle of magnetic waves and has a better broadband absorbing ability.

## 1. Introduction

With the rapid development of communication systems, stealth and anti-stealth technology have increasingly become decisive factors in military electronic information warfare. It is more and more essential to improve aircraft stealth. For single-station radar detection, the detection wave can be reflected to other angular domains through the shape structure design to achieve radar stealth. However, the expanding receiving angular territory for multi-station radar detection is far from enough to reflect the radar detection wave. Frequency Selective Rasorber (FSR) [1,2,3] directly absorbs the electromagnetic waves in the working frequency band, and the reflectivity is minimal. Therefore, FSR with a broadband wave absorption ability demonstrates an excellent application prospect.

FSR is a periodic array composed of metal patches or aperture units, which exhibit band-pass or band-stop spectral filtering characteristics for electromagnetic waves of different frequencies. It is widely used in radome [4], electromagnetic compatibility and electromagnetic shielding [5], satellite communication [6,7], and other fields. The traditional metal-dielectric Frequency Selective Surface (FSS) [8,9] is based on the scattering properties of metal resonator elements. When the electromagnetic wave is incidental on the surface of the FSS, an induced current is generated, thus generating a scattered field. The scattered and incident fields are superimposed to form an entire field with spatial filtering characteristics. This kind of FSS is generally a “sandwich” structure, which consists of a metal pattern on the top layer, a dielectric substrate in the middle, and a metal plate on the bottom layer, which is easy to process and low-cost. However, the disadvantage is that the absorption range is narrow, and the absorption rate is sensitive to the polarization mode and incidence angle of electromagnetic waves. Many scholars have researched and proposed various schemes to widen the absorbing bandwidth of FSS and reduce polarization sensitivity, such as designing a multi-mode resonant unit structure or fractal structure [10,11,12], using a multi-layer structure to realize the superposition of the absorbing frequency band [13] and loading lumped elements to realize bandwidth expansion [14]. They are adding active controllable devices such as PIN diode or varactor diode in the FSS [15,16] and using Spoof surface plasmon polaritons (SSPP) [17,18,19] to confine the incident wave to the interface and then dissipate it. In [20], the dispersive properties and subwavelength field confinement properties of artificial surfaces in low-frequency bands such as microwave and terahertz waves, which are similar to electrical surface plasmons in the optical frequency band, were studied for the first time. Since then, scholars have widely used SSPP in waveguide design [21], filters [22], notch filters [23], leaky-wave antennas [24,25], and other directions. At the same time, the designed FSR based on SSPP exhibits high rejection performance in the design stopband and achieves steep cutoff and high permeability in the working passband [26,27,28].

SSPP is based on Surface Plasmonic Polaritons (SPPs) by etching periodic structures on the metal surface or arranging periodic metal patches on the dielectric layer. When the electromagnetic wave is incident on the interface between the metal and the medium, the free electrons in the metal conductor oscillate collectively, generating SPPs in the microwave frequency band. The electromagnetic field strength peaks at the interface between the metal and the medium. The energy propagates along the structure’s surface and is completely bound near the structure’s surface, thereby achieving the wave-absorbing effect. Compared with SPPs, SSPP has more vital electromagnetic wave confinement ability, and the field strength decays exponentially in the vertical direction, reducing the interference to adjacent structures and being conducive to the miniaturized design of the FSR. In contrast, the unit structure of loading lumped components makes the FSR challenging to process and has poor practicability; if active devices are loaded, an external circuit needs to be introduced, which is slightly inconvenient. This paper uses a multi-layer stacking method to couple the SSPP to design the FSR.

## 2. Presentation of the Proposed Structure

The FSR designed in this paper is shown in Figure 1. It consists of a periodic square cavity array composed of a metal base plate and a multi-layer FSS cross-arranged. The depth of the square cavity unit is represented as *a*, and the length, width, and depth dimensions are consistent. The multi-layer FSS is shown in Figure 2a. The structure is symmetrically distributed concerning the indium tin oxide (ITO) layer. The dielectric is placed on the sub-top layer, and the sub-bottom layer is composed of polyresin (PR). The thickness of the dielectric layer is represented as *h*, and the dielectric constant is ε=4.3,tanδ=0.025. The ITO layer is placed in the middle, where the square resistance of ITO is represented as *R*, its unit is Ω/□, which characterizes the resistance value of a conductive material per square centimeter. Moreover, the thickness of ITO in the calculation model is 0. In reality, ITO is often formed by adhering to the surface of polyethylene glycol terephthalate (PET). Therefore, the ITO layer of the FSR is flanked by a PET layer with a thickness of *d*, and the dielectric constant of PET is ε=3.0,tanδ=0.18. The bottom and top layers are right-angle metal patch arrays. As shown in Figure 2b, the metal line width of the patch array is represented as *w*, the thickness is represented as *c*, and the distance between adjacent metal lines is also represented as *w*. These parameters are as follows: *R* = 200 Ω/□, a=10 mm, b=8 mm, c=35μm, d=10μm, w=125μm, and h=1 mm.

## 3. Design Principles

Compared with linear dipoles, the advantages of using L-shaped metal patches in this design are mainly reflected in two aspects. First, the same resonance characteristics can be obtained in both transverse electric (TE) and transverse magnetic (TM) modes. Second, one side of the L-shaped metal patch is equivalent to an inductor, and the other is equivalent to a capacitor between the adjacent patch. At the same time, the L-shaped metal patches array can achieve the most significant number with the most negligible length gradient so that the center frequency distribution of the FSS is tighter, which is more conducive to the realization of the effect of broadband wave absorption. The structure of the FSR designed in this paper is complex and cannot be directly analyzed as a whole utilizing an equivalent circuit. However, the FSR can be regarded as a multi-layer FSS with a cross-distribution in the xoz and yoz planes. According to the different incident angles of electromagnetic waves, there are two main mechanisms for the absorption principle of the FSR.

When the electromagnetic wave is vertically incident on the bottom surface of the absorber along the *z* direction, the wave vector direction is parallel to the wall surface of the square cavity. At this time, the wall surface of the FSR is equivalent to the SSPP transmission line. According to the SSPP theory, the electromagnetic wave in TE mode cannot induce polarization charges on the interface due to the absence of an electric field component perpendicular to the interface. Therefore, SSPP transmission lines only work in TM mode for specific surfaces. The FSR designed in this paper is a cross-distributed multi-layer FSS structure. For any set of mutually perpendicular multi-layer FSS, when the multi-layer FSS in the xoz plane is in TM mode, the multi-layer FSS in the yoz plane is in the TE mode, and vice versa. Therefore, relative to the whole FSR, whether it is TE mode or TM mode, the FSR can support SSPP. For a one-dimensional periodic groove array, when the groove width is much smaller than the incident wavelength (w≪λ), the dispersion relation can be expressed as Equation (Equation 1) [19]. Where $k_{z}$ is the wave number along the *z* direction of the conductor surface, the period of the groove array is represented as *D*, the width of the groove interval is represented as *A*, the depth of the groove is represented as *H*, and $k_{0}$ is the wave vector in the free space. Equation (Equation 2) is the electromagnetic wave’s attenuation constant along the wave vector’s propagation direction, which characterizes the attenuation ability of the transmission line to the electromagnetic field. From Equations (Equation 1) and (Equation 2), it can be concluded that in order to increase the attenuation capability of SSPP to electromagnetic waves, $k_{z}$ should be as large as possible so that the cut-off angular frequency ($w_{p}$) of SSPP can be obtained as shown in Equation (Equation 3). The FSR designed in this paper is no longer a simple one-dimensional groove array. Due to the reflection effect of the metal base plate and the non-negligible coupling effect between adjacent multi-layer FSS, its internal working mechanism will be more complicated.
(1)$${k_{z}}^{2}={k_{0}}^{2}\left(1+{\left(\frac{A}{D}\right)}^{2}\tan^{2}\left(k_{0}H\right)\right)$$
(2)$$\alpha_{T}=\sqrt{{k_{x}}^{2}-{k_{0}}^{2}}=k_{0}\frac{A}{D}\tan\left(k_{0}H\right)$$
(3)$$\omega_{p}=\frac{\pi c_{0}}{2H}$$

When the electromagnetic wave is vertically incident on the FSS wall of the FSR along the direction *x* or *y*, the wave vector direction is perpendicular to the wall of the square cavity. At this time, the working mechanism of the filter is similar to that of the traditional multi-layer FSS. The equivalent circuit of the formed FSS is shown in Figure 3. The upper and lower surfaces of the multi-layer FSS are air, and its impedance is $Z_{0}=377\;\Omega$. The PR layer with a thickness of *h* can be equivalent to a transmission line with a length of *h*, and its impedance is represented by $Z_{1}$, $Z_{1}$ = $Z_{0}/\sqrt{\varepsilon_{r}}$. The PET layer of thickness *d* can be equivalent to a transmission line of length *d*. Due to its small thickness, the impedance on the corresponding transmission line can be ignored. The ITO of the center layer is equivalent to a parallel resistance, which is represented by *R*. The FSS of the upper and lower layers is equivalent to two LC series branches, and the inductances of the branches interact with each other. Suppose the mutual inductance value and the impedance of the dielectric layer are ignored. In that case, the equivalent impedance of the multi-layer FSS can be obtained as in Equation (Equation 4). Where *L* is the equivalent inductance, *C* is the equivalent capacitance. The square resistance of the ITO layer is *R*, *R* = 200 Ω/□. The inductance corresponding to each metal patch in the FSS array is shown in Equation (Equation 5). Where *b* is half of the total length of the metal wire, *w* is the width of the metal wire, and $\mu_{0}$ is the vacuum permeability. The capacitance of the metal patch is shown in Equation (Equation 6), $\varepsilon_{0}$ is the vacuum conductivity, the effective dielectric constant of the dielectric layer is shown in Equation (Section 3) with $\varepsilon_{eff}$, and $\varepsilon_{r}$ is the relative dielectric constant of the medium. When the interaction between adjacent metal patches is not considered, it is not difficult to find the transmission pole corresponding to a single metal patch through Z=∞. By analogy, all the poles of the FSS can be found. The distance between adjacent metal patches is minimal, and their interaction cannot be ignored. At the same time, the FSR is a square cavity structure, and the interaction between adjacent walls and the reflection effect of the metal base plate cannot be ignored.
(4)$$Z=\frac{R\cdot \left(j\omega L_{2}+\frac{1}{j\omega C_{2}}\right)\cdot \left(j\omega L_{2}+\frac{1}{j\omega C_{2}}\right)}{R\cdot \left(j\omega L_{1}+\frac{1}{j\omega C_{1}}\right)+R\cdot \left(j\omega L_{2}+\frac{1}{j\omega C_{2}}\right)+\left(j\omega L_{1}+\frac{1}{j\omega C_{1}}\right)\cdot \left(j\omega L_{2}+\frac{1}{j\omega C_{2}}\right)}$$
(5)$$L=-\mu_{0}\frac{b}{\pi }\ln\left[\sin(\frac{\pi w}{4b})\right]$$
(6)$$C=-\varepsilon_{0}\varepsilon_{eff}\frac{4b}{\pi }\ln\left[\sin(\frac{\pi w}{4b})\right]$$
(7)εeff=1(εr+1)(εr+1)22

In fact, due to the complex structure of the FSR, the SSPP action mechanism and the traditional FSS action mechanism exist at any incident angle of electromagnetic waves, and the transmission characteristics of the absorber appear more complicated. In order to study the transmission characteristics of the absorber more accurately, we use the CST studio software to analyze the transmission characteristics of the whole waveband and finally make the FSR and test its transmission characteristics in the microwave anechoic chamber.

## 4. Analysis and Discussion

The working frequency band of the designed FSR should cover the S, C, X, Ka, and K bands as much as possible. Therefore, the cavity size of the FSR is optimized first, and then the ITO layer directly affects the circuit structure of the multilayer FSS. Therefore, the influence of the ITO square resistance on the transmission characteristics of the absorber is studied. The PR is a lossy medium, and the thickness of the dielectric layer also has a specific influence on the performance of the FSR. Finally, the optimization analysis of the width of the FSS metal patch is carried out. When the electromagnetic wave is vertically incident on the absorber along the *z* direction, the FSR is symmetrical about the xoz plane and the yoz plane. The transmission characteristics in the TE mode and TM mode are consistent, so only the transmission characteristics of the absorber in TE mode are studied. Figure 4 shows the absorptivity distribution of FSR with different sizes in the 0–30 GHz band when electromagnetic waves are incident vertically. It can be observed that when the incident frequency is less than 9 GHz, at the same frequency, with the decrease of *a*, the absorption rate of the FSR gradually decreases. When the incident frequency is greater than 9 GHz, the fluctuation characteristics of the absorption rate are enhanced. Except for the a=10 mm model, the absorption rates of other models have unstable fluctuations in different bands. When *a* is greater than 10 mm, the absorption rate gradually increases with the decrease of *a*. When the incident frequency is more than 18 GHz, the absorption rate remains unchanged with the decrease of *a*, and remains above 95%. In a comprehensive comparison, although the absorption rate of the a=10 mm model is lower than that of the a=15 mm and a=20 mm models when the incident frequency is less than 5GHz, the reduction is not too noticeable. However, the model with a=10 mm maintains the absorption rate above 95% in the 5–30 GHz band, which other models unmatch.

Figure 5 shows the absorption rate distribution of the model with the ITO layer and the model without the ITO layer in the 0–30 GHz band when the electromagnetic wave is vertically incident. It can be observed that when the incident frequency is less than 12 GHz, the FSR absorption rate is greatly improved due to the addition of the ITO layer. When the incident frequency is greater than 12 GHz, the ITO layer also dramatically reduces the fluctuation of the absorption rate of the FSR. In order to more accurately study the impact of the ITO layer on FSR’s inhalation transmission characteristics, this paper studies the distribution of FSR electric fields under the resonance frequency. Figure 6 shows the FSR electric field distribution when the incident frequency is 9 GHz. Due to the addition of the ITO layer, the strength of the electric field motivated on the FSR surface is significantly enhanced; this is mainly because the conductive characteristics of the ITO layer promote the coupling of the FSS on both sides of the square wall surface. To a certain extent, the electric field can penetrate the PR medium, thereby increasing FSS resonance intensity. This paper studies the effect of different square resistances of the ITO layer on the FSR performance. It can be observed from Figure 7 that, except for the *R* = 100 Ω/□ model, when the incident frequency is less than 5 GHz, the absorptivity gradually decreases with the increasing *R*. When the incident frequency is more than 5 GHz, except when the absorption rate of the *R* = 100 Ω/□ and *R* = 800 Ω/□ models decreases obviously, the changes in the absorption rates of other models are small.

Figure 8 shows the distribution of the influence of the thickness of the PR medium on the absorptivity. It can be observed that when the incident frequency is in the 3–7 GHz band, as *h* increases gradually, the absorptivity increases. When the incident frequency is in the 7–22 GHz band, the absorptivity is basically independent of *h*. When the incident frequency is in the 22–29 GHz band, the absorptivity of the h=0.8 mm model and h=0.6 mm model fluctuates wildly, and the absorption rate of the h=0.6 mm model drops to a minimum of 75%. Overall, the absorption rate increased gradually with the increase of *h* to more than 95%. However, when the incident frequency is more than 29 GHz, the absorptivity of the h=1.2 mm model drops to 80%, while the absorptivity of the h=1 mm model remains above 95%. After careful consideration, the PR medium thickness of the designed FSR is set to 1mm. This paper also studies the influence of the width of the FSS metal patch on the absorption rate. It can be observed from Figure 9 that the influence of the width of the metal patch within a specific range on the absorption rate is minimal. Except for the w=500 μm mode and w=250 μm model, the absorption rate is slightly lower than other models in some frequency bands, the absorption rate change is small, and both are above 90%. Considering the miniaturization design and processing difficulty of FSR, the w=125 μm model is finally selected.

Finally, the absorptivity of the FSR under different incident angles is compared and analyzed. In TE mode, as the incidence angle increases, the magnetic field direction always has an angle with the metal line on the FSR wall, and the component of the incident wave parallel to the metal line decreases with the increase in the incidence angle. The absorbing principle of the FSR designed in this paper is to use the SSPP to bind the spatial electromagnetic wave to the wall of the FSR, convert it into a plane wave, and finally dissipate it. As the incidence angle increases, the electromagnetic wave component bound to the wall of the FSR will become smaller, so the absorption rate of the FSR will decrease. It can be observed from Figure 10 that in the TE mode, with the increase in the incident angle of the electromagnetic wave, the most sensitive band to the incident angle is mainly in the 15–20 GHz band. When the incident angle increases to 20°, this band’s lowest point of absorptivity drop to 85%. When the incident angle increases to 40°, this band’s lowest point of absorptivity drops to 70%. The absorption rate in other bands is highly insensitive to the incident angle, and the absorption rate is above 90%. In TM mode, the direction of the magnetic field is parallel to part of the metal wire. As the incidence angle increases, the metal wire in the absorption cavity will be directly irradiated by the incident wave, thus increasing the utilization rate of the metal wire, so the absorption rate increases. Nevertheless, the incidence angle within a specific range has little influence on the absorption rate. It can be observed from Figure 11 that in the TM mode, when the incident angle increases to 40°, the lowest point of the absorption rate is up to around 95%.

## 5. Fabrication and Measurement

In order to verify the simulation results, the FSR based on SSPP is fabricated and processed using printed circuit board technology. The FSR size is 244mm×244mm×10mm, consisting 20×20 square cavity cell arrays, and the processed model is shown in Figure 12. The FSS on the inner wall of the square cavity is made by a printed circuit board. The ITO layer outside the square cavity is a dense ITO film formed on the PET substrate by magnetron sputtering under high vacuum conditions. The ITO-PET layer and the PR layer are tightly bonded together by hot pressing technology.

The experimental test adopts the stepped-frequency test system, and the schematic diagram of the system is shown in Figure 13. Agilent E8363A VNA (Agilent Technologies, Palo Alto, CA, USA) generates the stepped-frequency signal. Compared to point-frequency continuous test systems, the stepped-frequency test system does not require complex hardware cancellers. The signal sent by the test system is sent by the standard gain antenna aiming at the target, and the reflected echo signal is received by another standard gain antenna and sent to the vector network analyzer. Then, the vector network analyzer is used for time domain cancellation. Limited by the laboratory antenna specifications, the FSR is tested in the 1–18 GHz band, the test angle range is −40∼40°, and the test angle interval is 0.1°.

Figure 14 is the ISAR image of the metal plate and FSR. It can be observed that the FSR has a strong absorption of incident electromagnetic waves. Figure 15 compares simulated and measured results of FSR absorption rates in the TE mode under normal incidence. In the 1–5 GHz band, the absorption rate gradually increases and decreases with the increase of the incident frequency. At the same incident frequency, the absorption rate measured is slightly larger than the simulated results by about 10%. In the 5–8 GHz band, the absorption rate of FSR is basically about 90%, which is slightly smaller than the simulated results. In the 8–18 GHz band, FSR’s absorption rate is above 95%, consistent with the simulated results. Figure 16 compares the measured results of different incident angles in TE mode and the simulated results. It can be observed that with the increase in the incident angle of electromagnetic waves, the most sensitive bands to the incident angle are mainly in the 4.5–7 GHz band and the 13–18 GHz band. In these two bands, the absorptivity decreases roughly gradually with the increase of the incident angle. In the 4.5–7 GHz band, the fluctuation of the absorption rate is relatively small. When the incident angle increases to 40°, the lowest point of the absorption rate drops to around 77%. In the 13–18 GHz band, the absorptivity fluctuates wildly. When the incident angle increases to 20°, this band’s lowest point of absorptivity drop to around 83%. When the incident angle increases to 40°, this band’s lowest point of absorptivity drop to around 66%. Furthermore, absorptivity in other bands is less sensitive to the incident angle. Figure 17 compares the measured results of different incident angles in the TM mode and the simulated results. It can be observed that with the increase in the incident angle of electromagnetic waves, the absorption rate of FSR changed little, and the measured results are consistent with the simulation results.

By comparison, it is found that the experimental results are consistent with the numerical simulation results, proving that the FSR designed in this paper has excellent reliability.

Table 1 compares the proposed FSR with the reported FSR in terms of the working frequency band, band continuity, maximum incident angle, and size of the absorbing unit. The comparison shows that the proposed FSR has a wider absorption bandwidth, better band continuity, and weaker sensitivity to the incident angle. Furthermore, the smaller size of the absorbing unit, which is very conducive to the miniaturization design of FSR, makes the application prospect of FSR wider.

## 6. Conclusions

In this paper, a broadband FSR based on SSPP is designed, which is composed of a square cavity array composed of multiple layers of FSS. The numerical simulation results demonstrate that the designed FSR has an excellent absorption effect. When the electromagnetic wave is incident vertically, the absorption rate of the FSR in the 5–30 GHz band is above 95%, and in the 3.5–5 GHz band, the absorption rate is also above 80%. At the same time, FSR is highly insensitive to the incident angle. When the incident angle increases by 40°, the absorption rate fluctuates wildly only in the 15–20 GHz band, and the absorption rate drops to about 70% at the lowest. In TM mode, the incidence angle has a little effect on the absorption rate of FSR. The FSR designed in this paper is fabricated, and a microwave anechoic chamber measures its transmission characteristics. The experimental results demonstrate that the FSR absorption rate is consistent with the numerical simulation results, which proves that the FSR designed in this paper is less sensitive to the incident angle of magnetic waves. It has excellent broadband absorbing ability and outstanding reliability and robustness. The proposed FSR has a high absorption rate and little reflectivity of electromagnetic waves in its operating frequency band, so that it can be applied to the stealth design for dual-station radar detection. At the same time, the structural strength of the FSR is also excellent, which can realize the integrated design of structure and stealth and has a good application prospect in the stealth design of aircraft and other targets.

## Figures and Tables

**Figure 1 micromachines-13-01969-f001:**
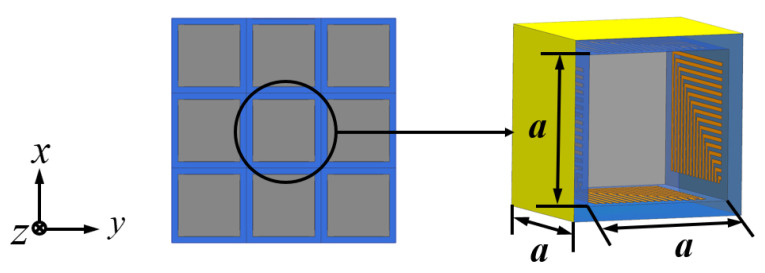
The structure of the FSR.

**Figure 2 micromachines-13-01969-f002:**
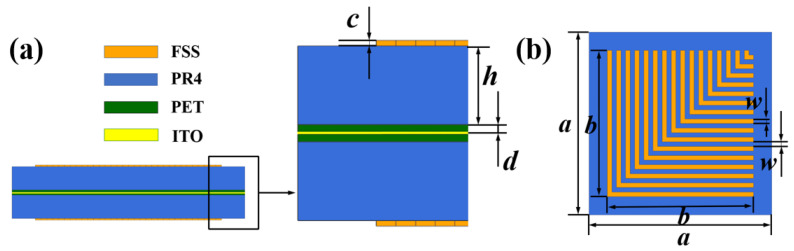
(**a**) Multi-layer FSS Sectional View. (**b**) The FSS patch array.

**Figure 3 micromachines-13-01969-f003:**
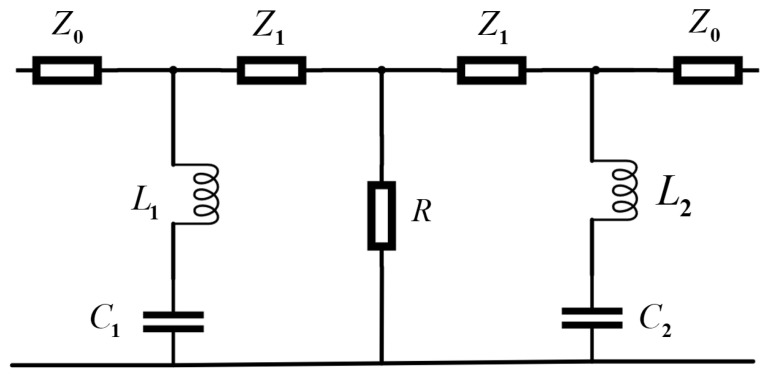
Equivalent circuit diagram of multi-layer FSS.

**Figure 4 micromachines-13-01969-f004:**
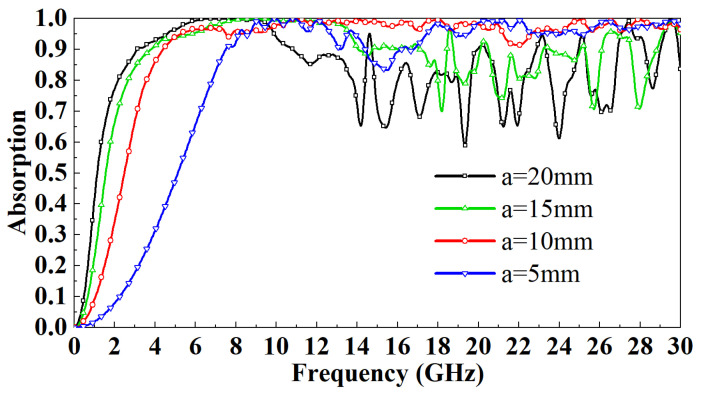
Effect of square cavity size on FSR absorption rate under normal incidence.

**Figure 5 micromachines-13-01969-f005:**
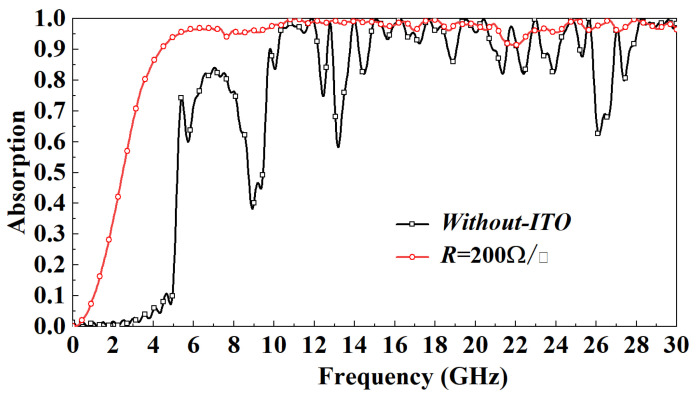
Effect of ITO layer on FSR absorption rate under normal incidence.

**Figure 6 micromachines-13-01969-f006:**
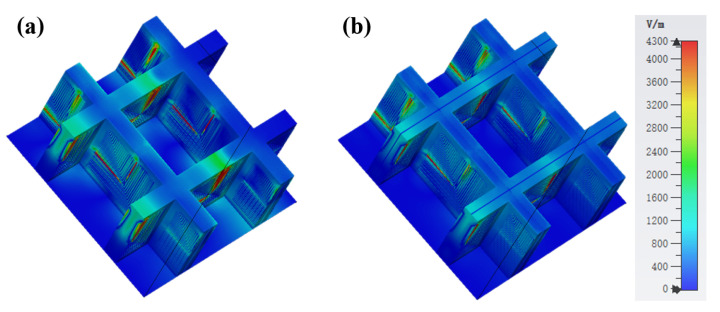
(**a**) *R* = 200 Ω/□ model’s E-field distribution at 9 GHz. (**b**) Without-ITO model’s E-field distribution at 9 GHz.

**Figure 7 micromachines-13-01969-f007:**
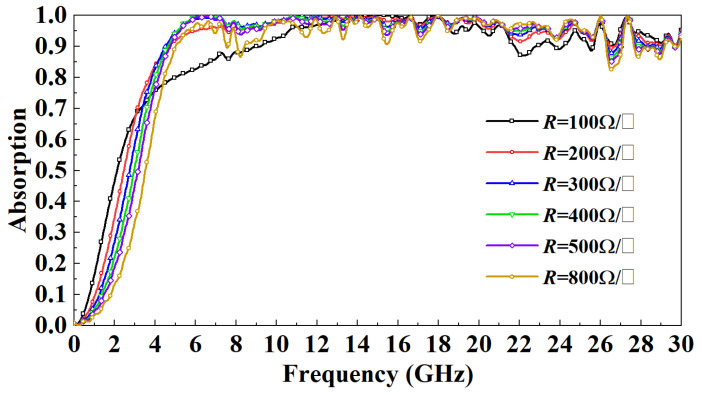
Effect of square resistance of the ITO layer on FSR absorption rate under normal incidence.

**Figure 8 micromachines-13-01969-f008:**
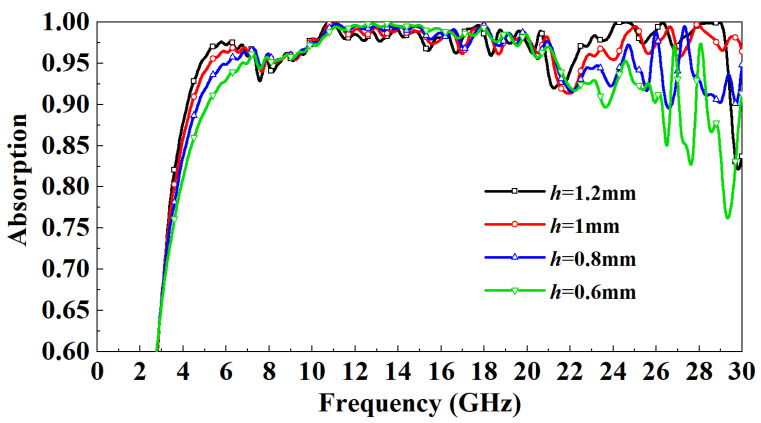
Effect of PR thickness on FSR absorption rate under normal incidence.

**Figure 9 micromachines-13-01969-f009:**
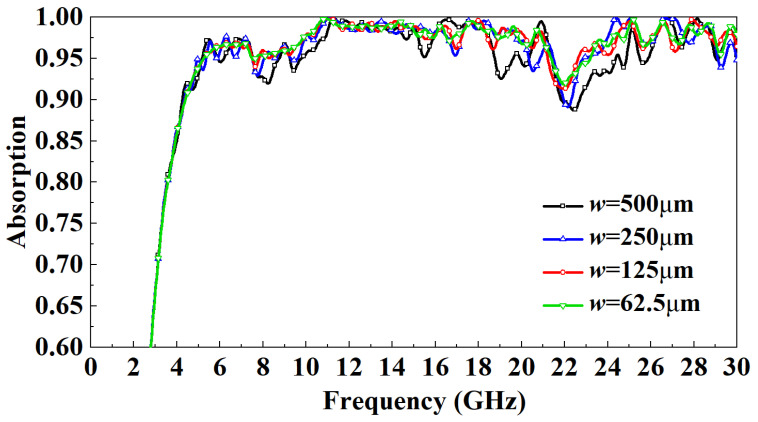
Effect of the width of the metal patch on FSR absorption rate under normal incidence.

**Figure 10 micromachines-13-01969-f010:**
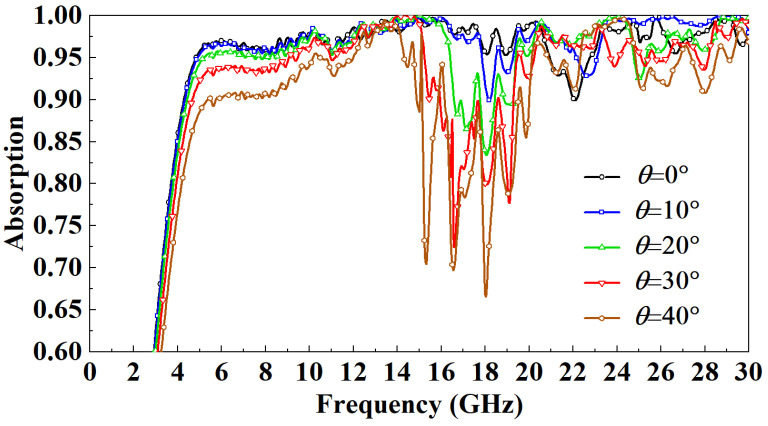
Effect of incident angle on FSR absorption rate in TE mode.

**Figure 11 micromachines-13-01969-f011:**
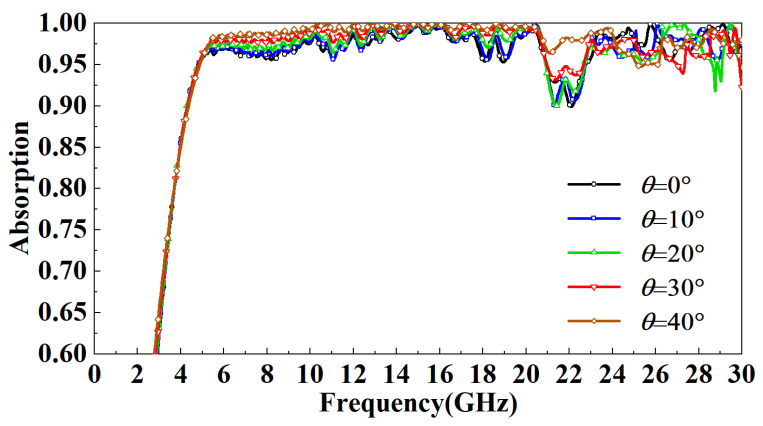
Effect of incident angle on FSR absorption rate in TM mode.

**Figure 12 micromachines-13-01969-f012:**
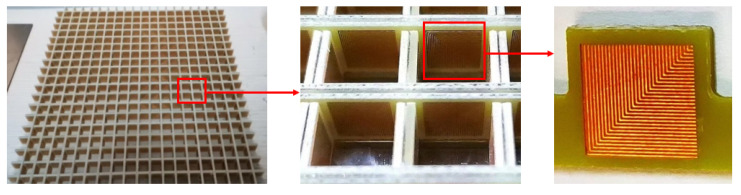
The photo of the FSR composed of 20×20 square cavity units, and the enlarged view of the FSS metal patch.

**Figure 13 micromachines-13-01969-f013:**
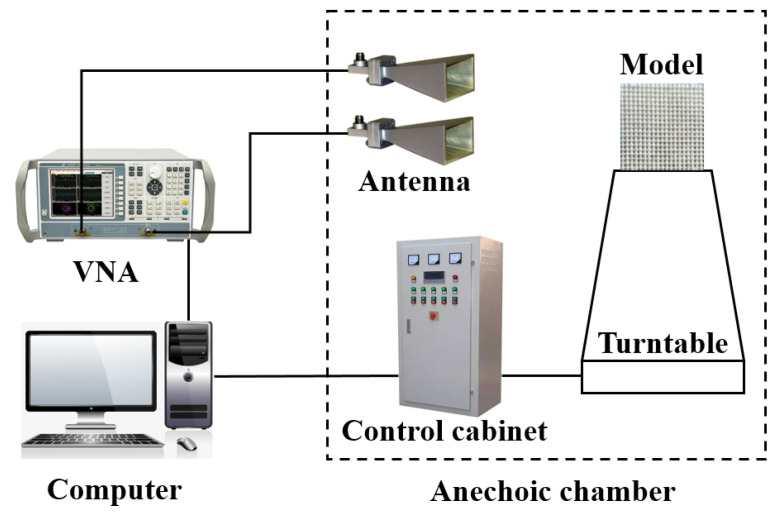
Stepped-frequency test system.

**Figure 14 micromachines-13-01969-f014:**
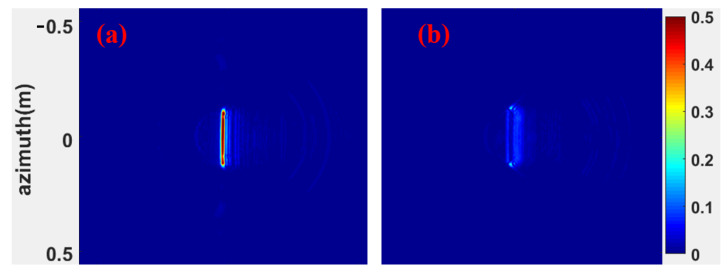
(**a**) ISAR imaging of sheet metal. (**b**) ISAR imaging of FSR.

**Figure 15 micromachines-13-01969-f015:**
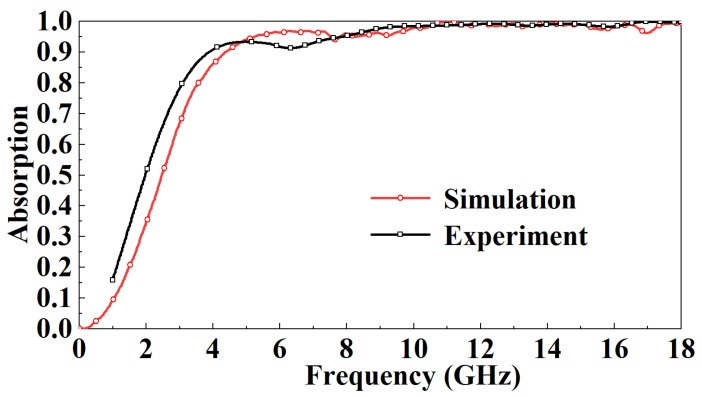
Comparison of simulated results and measured results of FSR absorption rate in TE mode under normal incidence.

**Figure 16 micromachines-13-01969-f016:**
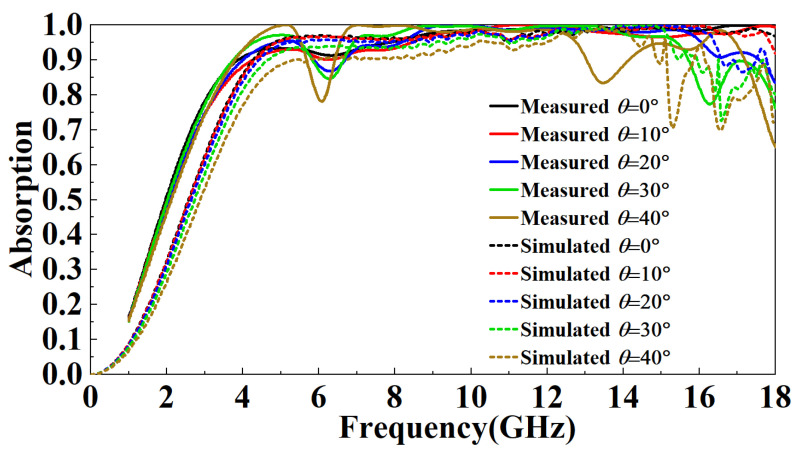
Comparison of the measured results of different incident angles in TE mode and the simulated results.

**Figure 17 micromachines-13-01969-f017:**
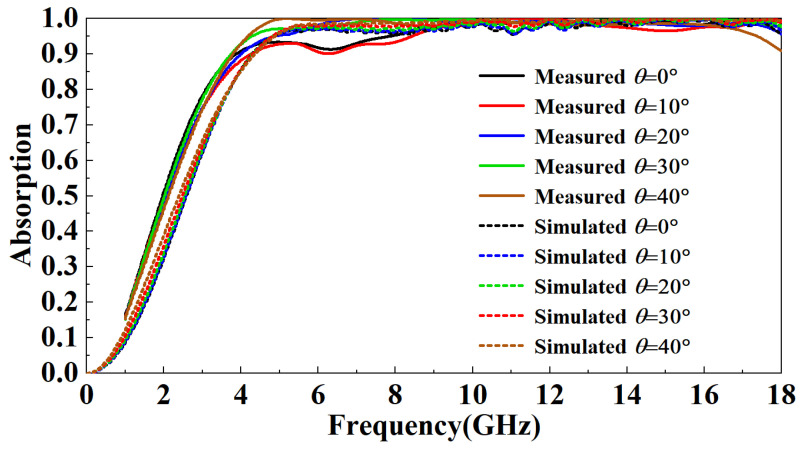
Comparison of the measured results of different incident angles in TM mode and the simulated results.

**Table 1 micromachines-13-01969-t001:** Feature comparison between the proposed FSR and the reported FSR.

FSR in Reference	Bandwidth/GHz	Band Continuity	Maximum Angle of Incidence	Cell Size/mm
[2]	7.99–11.97	Yes	Not reported	9 × 9 × 13.5
[3]	2.5–4.7; 7.8–14.6	No	40°	20 × 20 × 20
This work	3.5–30	Yes	40°	10 × 10 × 10

## Data Availability

The data that support the findings of this study are available from the corresponding author (Y.L.), upon reasonable request.

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
