# Peer review of "Broadband Frequency Selective Rasorber Based on Spoof Surface Plasmon Polaritons"

_micromachines, 2022, doi:10.3390/mi13111969_

Round 1

Reviewer 1 Report

The article is really well written and provides new scientific results. English language and style are fine, therefore the article is easy to read and understand the presented ideas and results.

Perhaps it should be explained in line 84 before Figure 1 in expression of units / what is , because it is not written in text.

The article is really well written, therefore we would like some comments in the conclusions, what could be the future works of this research.

Reviewer 2 Report

The paper presents a broadband FSS-based Rasorber. Interesting work and a sufficient amount of quality as well. I have some comments.

1. Only the main points included in the abstract

2. I suggest to simulate the equivalent circuit mentioned in Fig. 3 using ADS software and compare the simulated and measured results

3. In Fig 6, while changing from 100 ohms to 200 ohms, why good absorptivity is getting

4. Rasorber is a combination of Radome and absorber, but the paper not explained anything regarding radome

5. Explain theoretically, the working principle of absorber

Reviewer 3 Report

Authors have presented the work on “Broadband Frequency Selective Rasorber Based on Spoof Surface Plasmon Polaritons”. The work is presented nicely and very good absorption has been reported in broad band. Kindly find the thorough comments for the authors below;

1.       What is the novelty of this work compared to the state of the art? Author must include a comparison table with the state of the art.

2.       Correct GHz and other types in the figure captions in manuscript

3.       How does the author generate magnetic waves in anechoic chamber?

4.       Authors should include one graph to compare the TE and TM mode operations consistency.

5.       Authors should mention for the reason to choose Incident angles in the range of -40 to 40 degree not more?

6.       What is the reason of dropping the absorptivity as incidence angle increases?

7.       What type of the PCB substrate material is used to print the L-shaped printed lines? How its loss affect the performance of the FSR?

8.       How the rotation takes places for FSR measurement in the anechoic chamber? Vertical or horizontal? Is there any effect on the performance?

Reviewer 4 Report

Authors have presented work on “Broadband Frequency Selective Rasorber Based on Spoof Surface Plasmon Polaritons”. Absorption characteristics have been investigated under various condition and scenarios. The work is well presented. Following comments will be helpful to improve the presented work.

If possible, add reference for the equations for better understanding.

Check units in Figure 6 for the legends (ohm/??)

Add a comparison table or comments to further demonstrate and highlight the significance of the presented work.
